# Spin Susceptibility in Neutron Matter from Quantum Monte Carlo Calculations

**Luca Riz** [1,2,*] , **Francesco Pederiva** [1,2,*] , **Diego Lonardoni** [3,4] **and Stefano Gandolfi** [4]

1   Dipartimento di Fisica, University of Trento, via Sommarive 14, Povo, I–38123 Trento, Italy
2   INFN-TIFPA, Trento Institute for Fundamental Physics and Applications, I–38123 Trento, Italy
3   Facility for Rare Isotope Beams, Michigan State University, East Lansing, MI 48824, USA; lonardoni@nscl.msu.edu
4   Theoretical Division, Los Alamos National Laboratory, Los Alamos, NM 87545, USA; stefano@lanl.gov
*   Correspondence: luca.riz@alumni.unitn.it (L.R.); francesco.pederiva@unitn.it (F.P.)

**Abstract:** The spin susceptibility in pure neutron matter is computed from auxiliary field diffusion Monte Carlo calculations over a wide range of densities. The calculations are performed for different spin asymmetries, while using twist-averaged boundary conditions to reduce finite-size effects. The employed nuclear interactions include both the phenomenological Argonne AV8$'$ + UIX potential and local interactions that are derived from chiral effective field theory up to next-to-next-to-leading order.

**Keywords:** spin susceptibility; neutron matter; quantum Monte Carlo

## 1. Introduction

Spin susceptibility in neutron matter is known to be quite small [1–3]. The estimated values are of the order of $10^{-3}\,\mathrm{MeV}^{-1}\mathrm{fm}^{-3}$. This suggests that the magnetic fields that are commonly found in neutron stars are very unlikely to produce any significant deviation from the prediction of a model assuming strict zero spin polarization of the constituent matter. However, recent observations have revealed the existence of isolated neutron stars, the so called magnetars, where surface magnetic fields can reach intensities of the order of $\sim 10^{14}$–$10^{15}\,\mathrm{G}$ [4]. Moreover, in violent phenomena, like supernova explosions or neutron star mergers, the fluctuations of the magnetic field magnitude can reach very high peaks [5]. Therefore, general relativity simulations of such events might be sensitive to the details of the assumed values for the susceptibility [6–8]. For this reason, an accurate benchmark of the existing results is needed.

In this work, we present new results for the spin susceptibility in pure neutron matter (PNM) that is obtained from quantum Monte Carlo (QMC) calculations. We make use of the fact that it is possible to strongly reduce finite-size effects thanks to a modification of the system boundary conditions, overcoming some of the limitations of previous attempts to estimate the susceptibility with QMC methods [1]. In particular, we fix the external magnetic field and use the auxiliary field diffusion Monte Carlo (AFDMC) method [9,10] in order to study the energy per particle of the system as a function of the spin polarization, keeping the total number of particles fixed. We use twist-averaged boundary conditions (TABC) in order to reduce the finite-size effects [11].

Calculations are carried out for two different nuclear interactions that give realistic mass-radius relations for neutron stars. The first includes a phenomenological two- plus three-neutron potential, namely AV8$'$ + UIX. This interaction has been widely used in order to study light nuclei and neutron matter properties (see Refs. [12–14] and references therein). The second employs local potentials that are derived from chiral effective field theory (EFT) up to next-to-next-to-leading order (N$^2$LO) [15–19].

Among the different available formulations, we consider the interaction with coordinate-space cutoff $R_0 = 1.0$ fm and the $E\mathbb{1}$ parametrization of the three-body contact term $V_E$ (see Ref. [19] for details). Such a potential has been successfully used for nuclear structure and nuclear dynamics studies in nuclei [19–25], and it has been recently employed in order to derive the equation of state of nuclear matter and the symmetry energy, with good comparison with constraints from terrestrial experiments and multi-messenger astronomy [26].

The manuscript is organized, as follows. In Section 2, we provide a short overview of the periodic boundary conditions used in QMC calculations, and the technical details on the implementation of the TABC. In Section 3, we describe the procedures that were employed to estimate the spin susceptibility starting from the evaluation of the energy as a function of the strength of an external, static magnetic field. Section 4 contains the results of our AFDMC calculations, and Section 5 is devoted to conclusions.

## 2. Boundary Conditions in QMC

In QMC calculations, neutron matter is usually modelled as a system of $N$ particles in a box of fixed size $L$, so that the particle density is $\rho = N/L^3$. In order to avoid surface contributions, periodic boundary conditions (PBC) [10] are applied at the borders. This implies that the system is actually made up by an infinite number of boxes, periodically repeated, and all identical. From a dynamical point of view, this means that a particle exiting the box in a given direction re-enters on the opposite side coming from a neighbouring box. Wave functions are coherently chosen to this symmetry, and must be such that:

$$\psi\left(\mathbf{r}_1 + L\hat{\mathbf{x}}, \mathbf{r}_2, \dots\right) = \psi\left(\mathbf{r}_1, \mathbf{r}_2, \dots\right). \tag{1}$$

The ansatz for the PNM wave function used in QMC calculations consists of a symmetric Jastrow factor, encoding all of the information from the short-range correlations among particles, multiplied by a Slater determinant of plane waves with wave vectors compatible with the periodicity of the system [10].

The definition of the PBC is not univocal. In general, one can allow particles to pick up a phase $\theta$ when they wrap around the boundaries. This fact can be expressed by a more general formulation of the boundary conditions (here, for a wrap along the $x$ direction):

$$\psi\left(\mathbf{r}_1 + L\hat{\mathbf{x}}, \mathbf{r}_2, \dots\right) = e^{i\theta_x}\psi\left(\mathbf{r}_1, \mathbf{r}_2, \dots\right). \tag{2}$$

The extra phase $\theta_x$ is called a "twist" [11]. When the twist angle $\theta_x = 0$, Equation (2) yields back the standard PBC. Allowing for a generic twist angle $\theta_x \neq 0$ defines the TABC. Of course, it is possible to introduce separate twists for each system dimension, therefore defining a twist vector.

One of the shortcomings of the standard PBC applied to a homogeneous system of Fermions lays in the fact that the isotropy of the wave function in momentum space is lost. The average over randomly chosen twists helps to restore such a symmetry, thereby reducing the shell effects and kinetic energy finite-size errors [11].

Under TABC, physical observables need to be periodic in the twist angle, $F(\theta_i + 2\pi) = F(\theta_i)$. Because of this symmetry, the values on each component of the twist angle $\theta_i$ are restricted to the interval:

$$-\pi < \theta_i \leq \pi. \tag{3}$$

In order to respect the twisted boundary conditions, the wave vectors $\mathbf{k}_n$ of the plane waves must satisfy:

$$\mathbf{k}_n = (2\pi\mathbf{n} + \theta)/L, \tag{4}$$

where **n** is an integer vector.

The energy of each single particle state is given by $E_n = (\hbar^2/2m)\mathbf{k}_n^2$, and the ground state of the system is obtained by filling the lowest energy states. One should notice that the ordering of the single particle energies depends on the value of the twist [11]. The contributions of the twist angles to the energy decrease rapidly when increasing the number of particles, as expected from the notion of thermodynamical limit.

There are two aspects of using TABC that are extremely relevant for the evaluation of the spin susceptibility. First, it significantly reduces the number of particles that are necessary to describe the infinite system, at the cost of extra computation needed to sample the twist angles (much cheaper). Second, an arbitrary number of spin-up and spin-down particles can be used (in contrast to the PBC, where a number of particles corresponding to closed-shells has to be used), and thus a system with an arbitrary spin polarization can be described.

In Figure 1, we show results for the energy of PNM as a function of the number of neutrons $N$ while using PBC and TABC at saturation density $\rho_0 = 0.16\,\mathrm{fm}^{-3}$ (solid symbols). Alongside, we report the energies computed for the free Fermi gas (FG) (dotted-dashed and dashed curves). The results for the free FG are rescaled in order to match the PNM values as $\tilde{E}(N) = E(N) - E_\infty + E(66)$, where $E(N)$ is the energy per particle of the free system, $E_\infty$ is the correct result for the free FG at closed-shell configurations, and $E(66)$ is the energy of PNM with 66 particles. Brown triangles are the PBC results that are corrected for the kinetic energy of the FG, as done in Ref. [27].

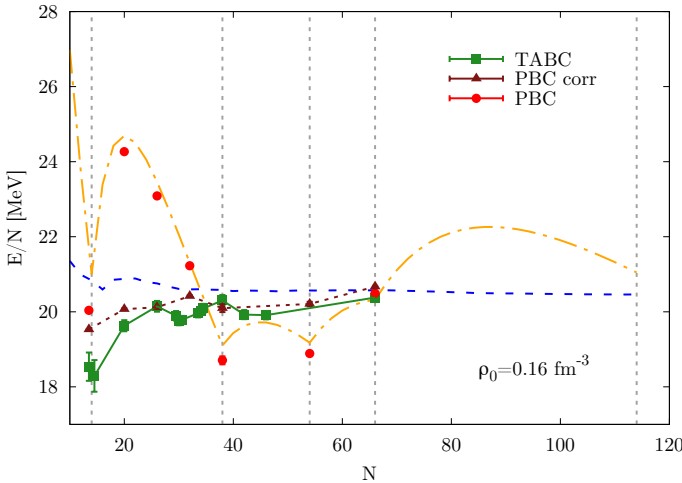

**Figure 1.** Pure neutron matter (PNM) energy results for the AV8′+UIX potential at saturation density using periodic boundary conditions (PBC) (red cirles) and twist-averaged boundary conditions (TABC) (green squares). Brown triangles are the results for PBC corrected for the Fermi gas (FG) kinetic energy (see text for details). Green solid and brown dotted lines connecting the TABC and PBC corrected results are just a guide to the eye. The FG solutions re-scaled to the interacting system are shown as orange dotted-dashed lines for PBC and as blue dashed lines for TABC.

The cusps in the energy curves computed while using PBC correspond to closed-shell configurations. The use of TABC strongly reduces shell effects: the convergence to the thermodynamic limit is much smoother, and a very reasonable approximation is already obtained for $N \geq 30$. In order to check the robustness of the calculations against a specific choice of the set of twist vectors, for selected cases we performed more calculations, sampling new twist vectors for each run. For clarity, the results for a given particle number, but different sets of twist vectors are plotted slightly shifted with respect to the correct particle number. Calculations were performed for $N = 14, 20, 26, 30, 34, 38, 42, 46, 66$. Because the free FG for $N = 66$ yields an energy that is particularly close to the thermodynamic limit, QMC calculations of PNM using PBC are usually carried out while using that particle number. Presently, AFDMC calculations with realistic nuclear interactions are limited to $N \sim 100$.

## 3. Computation of the Spin Susceptibility

The spin susceptibility quantifies the response of a system to the presence of a (static) external magnetic field. The Hamiltonian describing $N$ neutrons in the presence of a magnetic field can be written as:

$$H = H_0 - \sum_{i=1}^{N} \sigma_i \cdot \mathbf{b}, \tag{5}$$

where $\mathbf{b} = \mu\mathbf{B}$, with $\mu = 6.030774 \times 10^{-18}$ MeV/G. $H_0$ is the non-relativistic Hamiltonian of the interacting system with no external magnetic field, i.e., the sum of a non-relativistic kinetic term, and an interaction term that include two- and three-nucleon interactions [13].

The spin susceptibility is defined as:

$$\chi = -\rho\mu^2 \left. \frac{\partial^2 E_0(b)}{\partial b^2} \right|_{b=0}, \tag{6}$$

where $\rho$ is the density, and $E_0(b)$ is the ground-state energy per particle as a function of the magnetic field $b$. We use two different approaches in order to estimate the spin susceptibility:

i  Following the approximations of Ref. [1], the derivative in Equation (6) is estimated from an expansion of the energy as a function of the magnetization, computing the ground-state energy for a few selected values of field strength and spin polarization.
ii  Directly using the definition of the spin polarization $\xi$

$$\xi = \lim_{b \to 0} \xi(b), \tag{7}$$

where

$$\xi(b) = -\frac{\partial E_0(b)}{\partial b}, \tag{8}$$

the spin susceptibility can also be written as:

$$\chi = \lim_{b \to 0} \rho\mu^2 \frac{\partial \xi(b)}{\partial b}. \tag{9}$$

The derivative in the above equation is directly evaluated from the value of $\xi$ as a function of the magnetic field strength.

### 3.1. Use of the Pauli Expansion

Following Ref. [1], we can use the Pauli expansion of the energy as a function of the polarization $\xi$. From the expression of the energy per particle $E(b) = \langle H \rangle / N$, and remembering that

$$\xi = \frac{\langle \sum_{i=1}^{N} \sigma_i^z \rangle}{N}, \tag{10}$$

assuming that $\mathbf{b} = b\,\hat{\mathbf{k}}$, we obtain:

$$E(\xi) = E(0) - b\xi + \frac{1}{2}\xi^2 E''(0). \tag{11}$$

By minimizing $E(\xi)$ with respect to $\xi$, one can easily rewrite the definition of the spin susceptibility, as:

$$\chi = \mu^2 \rho \frac{1}{E''(0)}. \tag{12}$$

The quantity $E''(0)$ can be estimated by means of a few ground-state calculations for some fixed spin polarization $J_z = N_\uparrow - N_\downarrow$, where $N_\uparrow$ and $N_\downarrow$ correspond to Fermion closed-shells in a periodic box ($N_{\uparrow,\downarrow} = 1, 7, 19, 33, \ldots$), which can be easily realized within the AFDMC scheme. For any given $J_z$, we will have a different value of the energy as a function of the magnetic field $E(J_z, b)$, and consequently a different value of the spin polarization $\xi(J_z)$. By using the chain rule, it is possible to express $E''(0)$, as:

$$E''(0) = \left[ \frac{\partial \xi}{\partial J_z} \right]^{-2} \left\{ \frac{\partial^2 E_0}{\partial J_z^2} - \frac{\partial E_0}{\partial J_z} \left[ \frac{\partial \xi}{\partial J_z} \right]^{-1} \frac{\partial^2 \xi}{\partial J_z^2} \right\}. \tag{13}$$

By considering the ground-state energy ($\partial E_0 / \partial J_z = 0$), the above equation simplifies to:

$$E''(0) = \left[ \frac{\partial \xi}{\partial J_z} \right]^{-2} \frac{\partial^2 E_0}{\partial J_z^2}. \tag{14}$$

Assuming that:

1. for $b = 0$, $E_0(J_z, b)$ is quadratic in $J_z$ (see Figure 2);
2. for a fixed $J_z$, $E_0(J_z, b)$ is linear in $b$ (see Figure 3); and,
3. the spin polarization is linear in $J_z$,

the two derivatives in Equation (14) can be rewritten as:

$$\frac{\partial \xi}{\partial J_z} \approx \frac{E_0 \left( J_z = J_{z0}, b = 0 \right) - E_0 \left( J_z = J_{z0}, b = b_0 \right)}{J_{z0} b_0}, \tag{15}$$

and:

$$\frac{\partial^2 E_0}{\partial J_z^2} \approx 2 \frac{E_0 \left( J_z = J_{z0}, b = 0 \right) - E_0 \left( J_z = 0, b = 0 \right)}{J_{z0}^2}, \tag{16}$$

where $J_{z0}$ is the spin asymmetry of the ground state for a given external magnetic field $b_0$. These assumptions become exact in the limit of an infinite system with $J_z$ and $b$ small.

The energies entering Equations (15) and (16) can be directly computed while using the AFDMC method [10,26]. As previously mentioned, the main difference from the work of Ref. [1] is that, with TABC, we can access all spin polarizations with a fixed number of particles, while PBC limits the calculations to closed-shell configurations with roughly the same number of particles and only few spin polarizations can be analyzed. As a consequence, only very few values of spin polarization can be explicitly tested, when trying to determine the ground state. In order to overcome this limitation, the authors of Ref. [1] assumed the spin polarization to have the same energy dependence as in the Fermi gas. When using TABC, it is instead possible to compute the energy on a much finer grid of spin polarizations, allowing to explicitly search for its optimal value. The latter procedure is more accurate, and can, in principle, lead to results that are different from those that are found in Ref [1].

### 3.2. Use of the Spin Polarization

Another possibility, which also makes explicit use of TABC, is to try to directly determine, from AFDMC calculations, the ground-state polarization of the neutrons for a given value of the external magnetic field $b$. By interpolating the values end extracting the $b \to 0$ limit, it is possible to directly use the definition of Equation (9).

## 4. Results

We first verified that the assumptions yielding the approximate expressions of Equations (15) and (16) are reliable. In Figure 2, we report the results for the energy at saturation density for different values of $J_z/N$ and $b = 0$, while using $N = 38$ neutrons. The employed Hamiltonian includes the AV8'+UIX potential. The values are well described by a quadratic fit, at least for values $|J_z/N| < 0.2$. In Figure 3, the energy as a function of $b$ and for two different fixed values of $J_z$ is reported. Here, one expects the energy to be linear as a function of the strength of the magnetic field, and this behaviour is well confirmed.

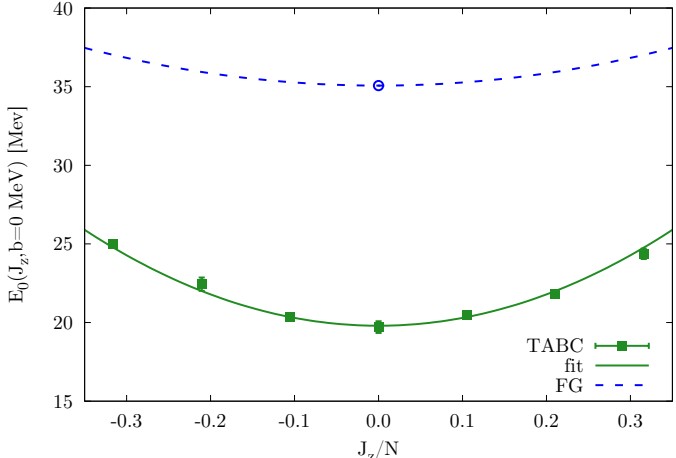

**Figure 2.** PNM energy per particle $E_0(J_z, b)$ at saturation density with seven different values of $J_z/N$ with no external magnetic field. Green squares with error bars indicate the AFDMC results with Monte Carlo statistical uncertainties while using TABC. The system is realized by 38 neutrons interacting via the AV8' + UIX potential, and five TABC are considered. The solid green line is a quadratic fit to the AFDMC results. The blue dashed curve shows the analytic solution of the free FG, with the ground-state polarization highlighted with a blue empty circle.

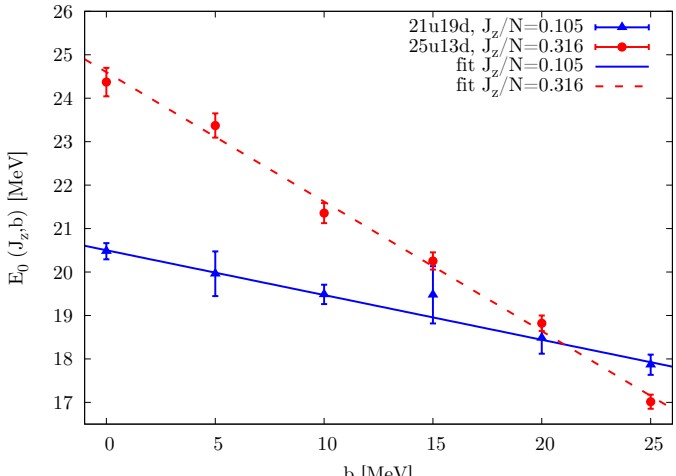

**Figure 3.** PNM energy per particle $E_0(J_z, b)$ at saturation density at fixed spin asymmetry $J_z$ in two different cases: 21 spin-up neutrons $N_\uparrow$ and 19 spin-down neutrons $N_\downarrow$ (blue triangles), and 25 $N_\uparrow$ and 13 $N_\downarrow$ (red circles), which correspond to $J_z/N = 0.105$ and $J_z/N = 0.316$, respectively. Solid and dashed lines are linear fits that show the validity of assumption (ii).

We then performed calculations of the energy per particle for seven different spin asymmetries for each external magnetic field. Fve different twist vectors have been randomly generated and

independent AFDMC calculations have been performed in order to implement TABC. The energy for each spin asymmetry is obtained by averaging over the different runs.

The value of $J_z/N$ at which the energy is minimized can be assumed as an estimate of the spin polarization of the system in a magnetic field $b$. Therefore, we performed a quadratic fit on the AFDMC results and determined the energy minimum. Such energy value is then used in Equations (15) and (16) to calculate the ground-state spin polarization $\xi(b)$. Uncertainties on the parameters, combined with the statistical errors from the Monte Carlo evaluation, provide the total uncertainty on the susceptibility.

In Figure 4, we report, as an example, the AFDMC results for PNM at saturation density with external field of 20 MeV and neutrons interacting via the AV8′ + UIX potential. Similar calculations have been performed at different densities, different external magnetic fields, and also for the local chiral EFT interaction. It is evident that the ground-state spin asymmetry of the interacting system is, in general, not coincident with the one predicted for a non-interacting FG at the same density.

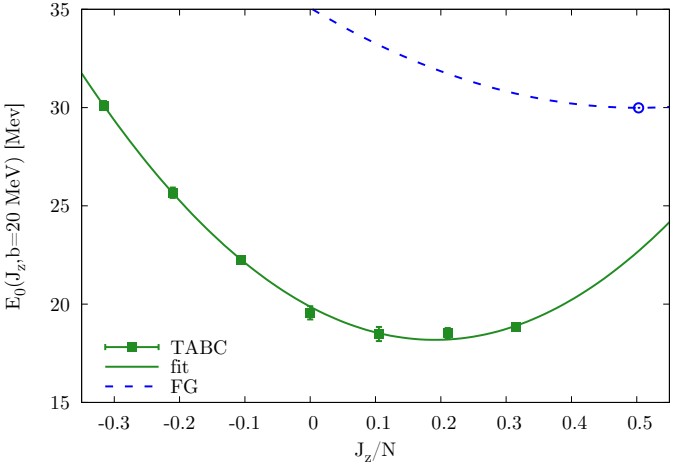

**Figure 4.** Same as Figure 2, but for external magnetic field $b = 20$ MeV.

The results of our QMC calculations are reported in Table 1. We obtain slightly different spin susceptibilities when compared to those from Fantoni et al. [1] for the same Hamiltonian, especially at low densities. This is mainly a consequence of performing our calculation at the correct ground-state spin polarization $\xi_0$ of the interacting system, rather than at the polarization that is predicted by the free FG. This is confirmed by the fact that, if we compute $\chi$ at the the same values of $\xi$ as in Ref. [1], then we obtain very similar predictions (see Table 1).

**Table 1.** Spin susceptibility ratio $\chi/\chi_F$ in PNM at different densities (in fm$^{-3}$). Results from Ref. [1] using the two-body potentials AV6′ and AV8′ supported by the three-body force UIX are reported in the second and third columns. The last two columns show the results of this work for the AV8′ + UIX potential. AV8′+UIX$^{(\star)}$ indicates the results obtained at the spin polarization of the ground state $\xi_0$ predicted while using the free FG, to be directly compared with the values reported in Ref. [1].

| | Ref. [1] | | This Work | |
| --- | --- | --- | --- | --- |
| $\rho$ | AV6′ + UIX | AV8′ + UIX | AV8′ + UIX$^{(\star)}$ | AV8′ + UIX |
| 0.08 | | | 0.40(2) | 0.45(2) |
| 0.12 | 0.40(1) | | 0.42(3) | 0.45(3) |
| 0.16 | | | 0.39(2) | 0.33(2) |
| 0.20 | 0.37(1) | 0.39(1) | 0.36(2) | 0.38(1) |
| 0.32 | 0.33(1) | 0.35(1) | 0.34(2) | 0.31(1) |
| 0.40 | 0.30(1) | | 0.29(2) | 0.30(1) |

The ground-state spin polarizations computed with and without interaction are significantly different at all densities $\rho$ and at all external magnetic fields $b$, as shown in Figure 5. It can be noticed that, also for the interacting case, the ground-state polarization is always a decreasing function of the density, coherently with the predictions of the FG theory.

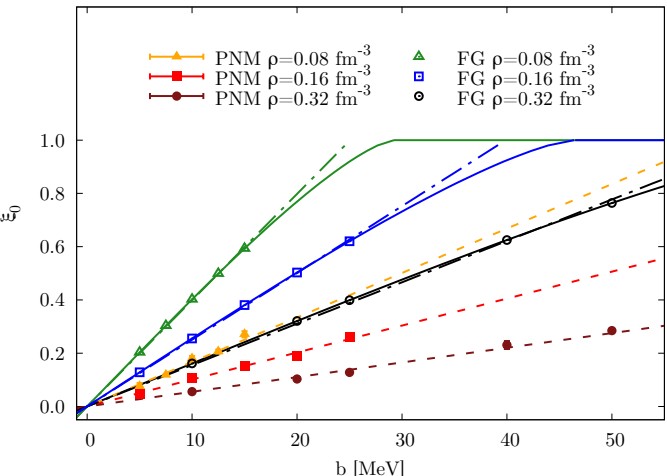

**Figure 5.** Ground-state polarization $\zeta_0$ as a function of the external magnetic field $b$. The results for PNM with the AV8′ + UIX potential (solid symbols) are compared to those for the free FG (empty symbols) at three different densities. Analytic results for the FG are shown with solid lines, while linear fits to some mock data (generated from the analytic solution of the free FG at five different external magnetic fields $b$) are shown with dotted-dashed lines. The results for the interacting system obtained from the quadratic fits of approach (i) are shown with orange-red-brown (from low to high density) dashed lines.

The second approach that is mentioned in Section 3 directly relies on the definitions of the spin susceptibility and the spin polarization. As an example, we report, in Figure 5, the results of the ground-state spin polarization $\zeta_0$ as a function of the external magnetic field $b$ for PNM with AV8′ + UIX at three different densities. Similar results can also obtained for local chiral EFT potentials, and at different densities.

In Figure 6, we report the estimates of the spin susceptibility in PNM with neutrons interacting with both the phenomenological AV8′ + UIX potential and the local chiral EFT interaction. A comparison between the two approaches discussed in the previous section for the AV8′ + UIX case is also reported in the same figure. The direct estimation of the susceptibility gives a smoother dependence on the density. This fact might be related to the absence of residual systematic errors that are still likely to plague the indirect estimate described above. For as concerns the results from the EFT Hamiltonian, they do not significantly differ from those that were obtained from a phenomenological interaction over all the density range considered.

In order to estimate the uncertainty due to the assumption of a linear dependence of the energy on the magnetic field $b$, we can once again resort to the FG analysis. In fact, in the case of the non-interacting system, we can compute the analytic solution and analyze the discrepancies with the results obtained under the assumption of a linear dependence. The linear approximation yields to slightly underestimated values for the susceptibility, in particular at larger densities, as shown in Figure 7. In principle, there is no reason to assume a different behaviour for an interacting system, although the correct result is unknown.

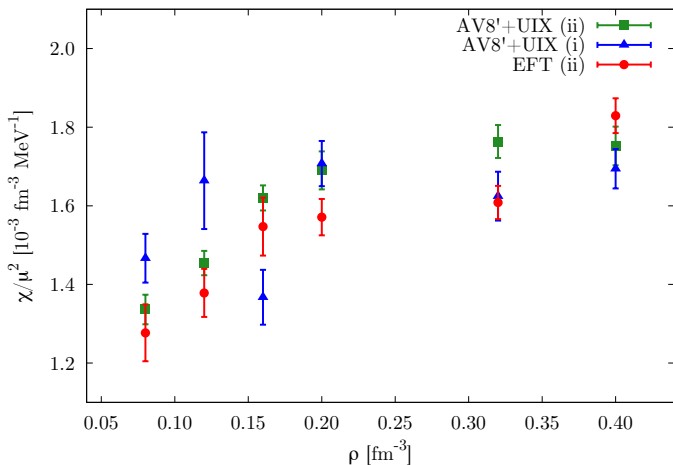

**Figure 6.** Spin susceptibility $\chi$ in PNM for the AV8$'$ + UIX (green squares) and the local chiral EFT (red circles) potentials as a function of the density from approach (ii). Results for AV8$'$ + UIX from approach (i) are shown with blue triangles. Uncertainties on the fitting parameters have been propagated and they are shown by the error bars.

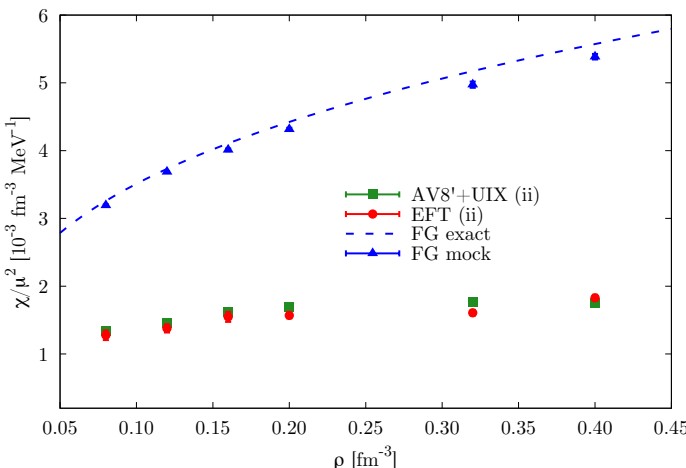

**Figure 7.** Same as Figure 6 but results are shown for PNM from approach (ii), and for the free FG from analytic solution (blue dashed line) and based on mock data (blue triangles), see Figure 5.

In Figure 8, we compare our results for the spin susceptibility to those available from Brueckner–Hartree Fock (BHF) calculations. In black, we report the calculations by Bombaci et al. [3] for the two-body AV18 potential. The results of Vidaña et al. [2], obtained using the two-body NSC97e interaction in a parametrization for high-spin polarizations, are shown with the orange band. The results look qualitatively consistent overall, even though a fair comparison is difficult due to the different employed nuclear potentials, many-body methods, and strategy used to estimate the spin susceptibility. For phenomenological potentials, the effect of the three-body force appears to be non negligible, as shown by the $\approx$10–15% difference between BHF AV18 and AFDMC AV8$'$ + UIX results. Realistic interaction models, including both two- and three-body forces (AV8$'$ + UIX and local chiral EFT), appear to provide consistent susceptibilities, regardless the scheme of the interaction.

Finally, we analyze an alternative procedure of computing the spin susceptibility that only relies on the equation of state (EOS) of spin unpolarized PNM and fully spin polarized PNM. We first define an energy density functional $\varepsilon$ of the form:

$$\varepsilon = \varepsilon_0(\rho) + \xi^2 \left[\varepsilon_1(\rho) - \varepsilon_0(\rho)\right], \tag{17}$$

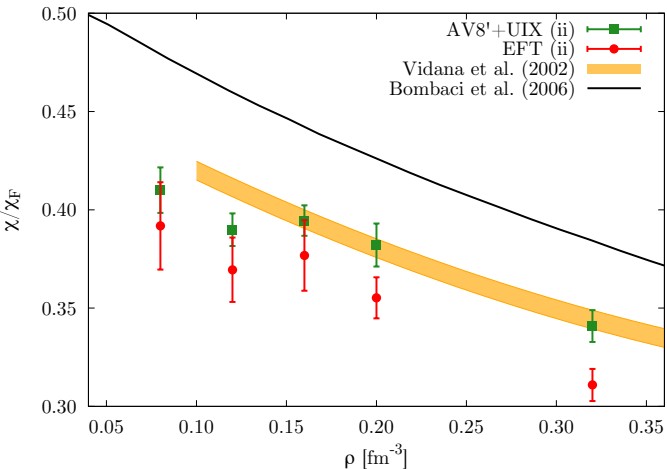

**Figure 8.** Spin susceptibility ratio $\chi/\chi_F$ in PNM as a function of the density. Green squares (red circles) are results for the AV8′+UIX (local chiral EFT) potential from approach (ii). The black line shows the BHF results of Ref. [3] for the two-body interaction AV18. The orange band reports the BHF results of Ref. [2] for the two-body NSC97e interaction in a parametrization for high-spin polarizations.

where $\varepsilon_0(\rho)$ and $\varepsilon_1(\rho)$ are two parametrizations with the same functional form for the spin unpolarized and fully spin polarized system, respectively. The quadratic dependence of the energy as a function of the spin polarization $\xi$ is a common assumption that is derived from the expansion of the energy of the partially polarized FG. By inserting the energy function $\varepsilon$ in the definition of Equation (6), we obtain that the spin susceptibility can be calculated as:

$$\chi = \mu^2 \rho \, \frac{1}{2\left[\varepsilon_1(\rho) - \varepsilon_0(\rho)\right]}, \tag{18}$$

where we can define the spin-symmetry energy as $E_{\text{sym}}^\sigma(\rho) = \varepsilon_1(\rho) - \varepsilon_0(\rho)$.

This holds for any expression of $\varepsilon_i(\rho)$. Using AFDMC calculations for spin unpolarized PNM (from Ref. [27] for the AV8′ + UIX potential, and from Ref. [18] for the local chiral EFT interaction) and fully spin polarized PNM (from Ref. [28]), we directly compute the spin-symmetry energy, as shown in Figure 9.

While using Equation (18), we can directly compute the spin susceptibility. The results are shown in Figure 10 and they are compared to the values obtained using the approach (ii). The spin susceptibility for the phenomenological potential provides a better agreement between the two approaches, while the comparison is worse for the local chiral EFT interaction.

This approach, although really simple, provides a reasonable estimate for the spin susceptibility in PNM. In contrast, the free FG solution predicts a spin susceptibility larger by a factor $\approx 3$ than the one that was obtained for the interacting case.

The spin polarization of neutron matter can be affected by the presence of strong magnetic fields in neutron stars. However, the magnetic fields that are needed to significantly spin polarize neutron matter are very strong. The observed magnetic field at the neutron star surface is $B \approx 10^{11}$ T. Using the simple model $2\xi\Delta E = \mu_n B$, where $\Delta E$ is the spin-symmetry energy and $\mu_n$ the chemical potential, at saturation density the estimated value of the spin polarization is $\xi \approx 10^{-4}$.

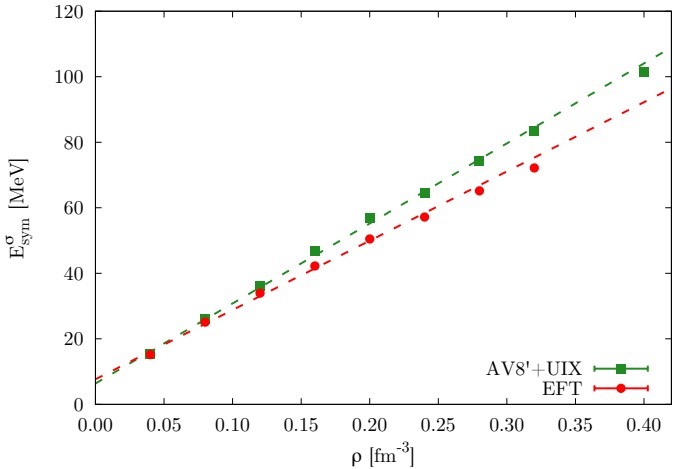

**Figure 9.** AFDMC results for the spin symmetry energies. Statistical Monte Carlo uncertainties are smaller than the symbols. The dashed lines are linear fit to to the AFDMC results.

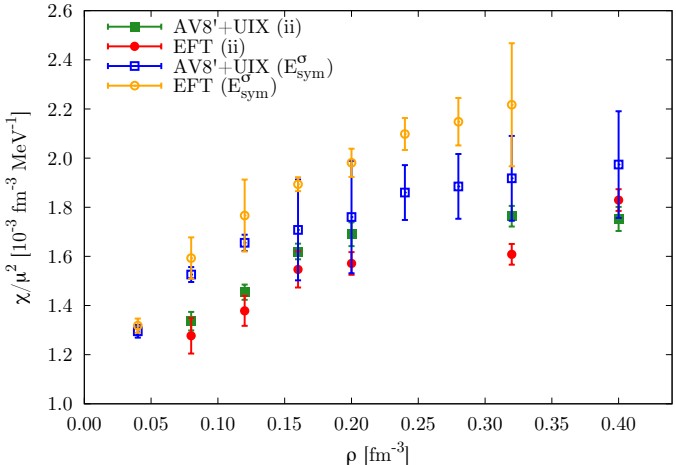

**Figure 10.** Same as Figure 6 but results are shown for PNM from approach (ii) and from $E_{sym}^{\sigma}$.

## 5. Conclusions

We performed a substantially improved AFDMC calculation of the spin susceptibility in PNM. The main result is that the predicted ground-state polarization of the interacting system is lower than the one that is predicted by the free FG. Although previous calculations did not take such an effect into account, this does not lead to significant changes in the magnitude of the spin susceptibility. Implementing TABC allowed us to check the reliability of the assumptions made in previous works [1]. Moreover, the spin susceptibility is now calculated more consistently: on the one hand, we can perform calculations with an arbitrary number of particles, therefore significantly reducing finite-size effects; on the other hand, we can perform calculations with arbitrary spin polarization, keeping the total number of particles fixed. The same technique can also be directly extended to isospin-asymmetric matter. We performed and compared calculations while using both the phenomenological potential AV8′ + UIX and a local chiral EFT interaction up to N²LO. We presented a method, namely approach (ii), in order to directly compute the spin susceptibility from the energy of a system subject to an external magnetic field, which relies on the possibility of performing AFDMC calculations for arbitrary number of particles in a periodic box, minimizing the finite size effects by means of TABC. This approach has an analogous counterpart for the free FG, which provides a reasonable way of estimating uncertainties.

**Author Contributions:** Conceptualization, F.P.; Data curation, L.R. and S.G.; Methodology, D.L.; Supervision, F.P. and S.G.; Writing—original draft, L.R. and F.P.; Writing—review & editing, L.R., F.P., D.L. and S.G. All authors have read and agreed to the published version of the manuscript.

**Funding:** The work of D.L. was supported by the U.S. Department of Energy, Office of Science, Office of Nuclear Physics, under the FRIB Theory Alliance award DE-SC0013617, and by the NUCLEI SciDAC Program. The work of S.G. was supported by U.S. Department of Energy, Office of Science, Office of Nuclear Physics, under Contract No. DE-AC52-06NA25396, by the DOE NUCLEI SciDAC Program, by the LANL LDRD Program, and by the DOE Early Career Research Program. Computational resources have been provided by CINECA through the INFN computing time allocation to the MANYBODY National Project, and by the National Energy Research Scientific Computing Center (NERSC), which is supported by the U.S. Department of Energy, Office of Science, under Contract No. DE-AC02-05CH11231.

**Acknowledgments:** We thank Kevin Schmidt and Albino Perego for useful discussions and comments.

**Conflicts of Interest:** The authors declare no competing interests.

## Abbreviations

The following abbreviations are used in this manuscript:

| | |
|---|---|
| PNM | pure neutron matter |
| QMC | quantum Monte Carlo |
| AFDMC | auxiliary field diffusion Monte Carlo |
| TABC | twist-averaged boundary conditions |
| EFT | effective field theory |
| $N^2LO$ | next-to-next-to-leading order |
| PBC | periodic boundary conditions |
| FG | Fermi gas |
| BHF | Brueckner-Hartree Fock |

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
