# Peer review of "Spin Susceptibility in Neutron Matter from Quantum Monte Carlo Calculations"

_2571-712X, doi:10.3390/particles3040046_

Round 1

Reviewer 1 Report

In this manuscript, the authors have studied the spin polarization and the spin susceptibility of pure neutron matter using twist-averaged boundary conditions, by employing the Argonne AV8'+UIX potential and the nucleon interaction from the chiral effective field theory. The results are compared with that from a previous auxiliary field diffusion Monte Carlo calculation as well as those from Brueckner-Hartree Fock calculations. I think this is a solid study with results well presented, and worth publication. I have the following questions/comments and suggested changes before accepted for publication.
1) At the end of Sec.3.1 the authors claimed that the main difference between their calculation and Ref.[1] is from the different boundary conditions, while above Table.1 the authors mentioned that the difference from Ref.[1] can also be using a different ground state. In addition, statements above Table. 1 seem to say that their spin susceptibility is significantly smaller than that from Ref.[1] due to the different ground states used. This seems to be only true for \rho=0.32 fm^-3 but not for other densities. I think the authors should clarify more clearly the difference between their calculation and Ref.[1], and revise the statements above Table.1.
2) In Fig.8, the curve by Bombaci is significantly higher than that from the present study. This is attributed to the different three-body interactions in the manuscript. I think the reason should be further clarified with a deeper discussion.
3) In Fig.5, I suggest that the maximum scale of the y axis is set to be 1.
4) It is interesting that the authors discussed the spin susceptibility and even the spin symmetry energy at different densities. The authors are encouraged to relate their results in nuclear matter to observational consequences in neutron stars under strong magnetic fields, as mentioned in the introduction.

Author Response

We thank the Referee for carefully reading the manuscript and for providing constructive comments. Below is the detailed response to the Referee's report.

1. The work presented in this manuscript is obtained by using the twist-averaged boundary conditions (TABC), while that of Ref.[1] used the simpler periodic boundary conditions (PBC). In the latter, calculations of systems in presence of an external magnetic field, with non-zero ground-state spin-polarization, can be reliable only in the fortuitous case in which the ground-state polarization coincides with a value that can obtained using number of spin-up and spin-down neutrons corresponding to closed shells in a box. In general, an interpolation constrained in a very flimsy way must be performed to find the ground-state. In Ref.[1] the authors were indeed not able to compute the correct ground-state polarization directly from PBC calculations, and they assumed that it would follow that of the Fermi gas. Even though the interpolation of PBC results can be theoretically carried out, such a fit is way more approximated than using the TABC. For these reasons, in the manuscript we added a few sentences stating that the ground-state inferred from TABC or from PBC calculations can be different, hopefully clarifying this aspect. We thank the Referee for pointing out the statement above Table 1, that was, in fact, a bit misleading. We also improved this statement in the amended version of the manuscript.

2. In Fig.8, the curve by Bombaci et al. has been obtained using the two-body interaction AV18 alone. No three-body force was involved in that calculation. However, it is know (see for instance Ref.[13] and reference therein) that for phenomenological potentials such as AV18 the effect of three-body forces in PNM is large, especially in dense matter, and cannot be neglected. The results presented in this work have been obtained using realistic Hamiltonians, both derived phenomenologically and from chiral EFT, explicitly including two- and three-body interactions. Different is the case of the orange band from Vidaña et al., obtained using a meson-exchange two-body potential only. This, by considering different mesons with mass higher than the pion one, effectively includes contributions that will correspond to a three-body force in the phenomenological scheme. Hence, the orange band is closer to the results using two- plus three-body potentials. However, as already pointed out by the Authors in the manuscript, it has to be noted that the results in Fig.8 have also been obtained using different many-body methods and strategy to estimate the spin susceptibility, and therefore a fair comparison is difficult. A deeper discussion of the specific differences between the different approaches is in any case outside the scope of the current work.

3. The Authors agree with the Referee regarding the maximum value of the y axis in Fig.5. We fixed it at 1.0 and used the remainder space above for the legend.

4. We agree with the Referee that making deeper connections with neutron stars would be very useful. However, the magnetic fields needed to spin polarize neutron matter are very strong. The observed magnetic field at the neutron star surface is $B\approx10^{11}T$. Using the simple model $2\xi\DeltaE=\mu_n B$, where $\Delta E$ is spin-symmetry energy and $\mu_n$ the chemical potential, at saturation density we get $\xi\approx10^{-4}$. Stronger magnetic fields might not be ruled out in the interior. However, the many different models existing in the literature would require a discussion that is more appropriate for a separate paper. We added this statement in the manuscript.

Reviewer 2 Report

The paper addresses spin susceptibility of pure neutron matter (PNM). This problem is important in the nuclear physics and physics of neutron stars. There were several studies of this problem previously, but the present work treats it at a new, modern level, including a modern interaction model and using quantum Monte Carlo numerical simulations.

The new interaction model is the well-known N2LO approximation in the chiral effective field theory, developed in the last decade. In addition, the authors also consider an alternative, more phenomenological model of neutron interactions, AV8'+UIX, which was previously employed in a number of studies of the PNM. They compare the results of these two models and additionally the results obtained from the analytic Fermi gas model. In Fig.8 they additionally make a comparison with two other previously published results, based on other models of the interaction and theoretical methods. The comparison shows a reasonable agreement of these different results.

Although the nucleon spin susceptibility is known to be small at the considered densities (within about a factor of 2 around the nuclear saturation density), the results are important, since they give the quantitative evaluation of this smallness, which may be useful in the studies of ultramagnetized objects, such as magnetars, neutron-star mergers, and some models of proto-neutron stars. It is interesting to see that the spin susceptibility of the PNM in the model with interactions is a few times smaller than it would be in the ideal Fermi gas model.

The paper is very well written, all methods and assumptions are clearly described, the presentation of the results and the conclusions are clear and sound.

I believe that this paper can be published in the present form.

Author Response

We thank the Referee for the very positive feedback on the manuscript and for suggesting publication in the present form.